# Chronic N-Acetylcysteine Treatment Prevents Amphetamine-Induced Hyperactivity in Heterozygous *Disc1* Mutant Mice, a Putative Prodromal Schizophrenia Animal Model

**DOI:** 10.3390/ijms23169419

**Published:** 2022-08-20

**Authors:** Chuan-Ching Lai, Rathinasamy Baskaran, Chih-Yu Tsao, Li-Heng Tuan, Pei-Fen Siow, Mahalakshmi Palani, Lukas Jyuhn-Hsiarn Lee, Chih-Min Liu, Hai-Gwo Hwu, Li-Jen Lee

**Affiliations:** 1Department of Post-Baccalaureate Veterinary Medicine, Asia University, Taichung 413305, Taiwan; 2Department of Bioinformatics and Medical Engineering, Asia University, Taichung 413305, Taiwan; 3Graduate Institute of Anatomy and Cell Biology, National Taiwan University College of Medicine, Taipei 100233, Taiwan; 4School of Medicine, National Tsing Hua University, Hsinchu 300044, Taiwan; 5Institute of Systems Neuroscience, National Tsing Hua University, Hsinchu 300044, Taiwan; 6Department of Medical Science, National Tsing Hua University, Hsinchu 300044, Taiwan; 7Institute of Environmental Health Sciences, National Health Research Institutes, Miaoli 350401, Taiwan; 8Department of Psychiatry, National Taiwan University Hospital, National Taiwan University College of Medicine, Taipei 100233, Taiwan; 9Neurobiology and Cognitive Science Center, National Taiwan University, Taipei 100233, Taiwan; 10Institute of Brain and Mind Sciences, National Taiwan University College of Medicine, Taipei 100233, Taiwan

**Keywords:** schizophrenia, prodromal phase, animal model, dopamine receptor, striatum

## Abstract

Symptoms of schizophrenia (SZ) typically emerge during adolescence to young adulthood, which gives a window before full-blown psychosis for early intervention. Strategies for preventing the conversion from the prodromal phase to the psychotic phase are warranted. Heterozygous (Het) *Disc1* mutant mice are considered a prodromal model of SZ, suitable for studying psychotic conversion. We evaluated the preventive effect of chronic N-acetylcysteine (NAC) administration, covering the prenatal era to adulthood, on the reaction following the Amph challenge, which mimics the outbreak or conversion of psychosis, in adult Het *Disc1* mice. Biochemical and morphological features were examined in the striatum of NAC-treated mice. Chronic NAC treatment normalized the Amph-induced activity in the Het *Disc1* mice. Furthermore, the striatal phenotypes of Het *Disc1* mice were rescued by NAC including dopamine receptors, the expression of GSK3s, MSN dendritic impairments, and striatal PV density. The current study demonstrated a potent preventive effect of chronic NAC treatment in *Disc1* Het mice on the acute Amph test, which mimics the outbreak of psychosis. Our findings not only support the benefit of NAC as a dietary supplement for SZ prodromes, but also advance our knowledge of striatal dopamine receptors, PV neurons, and GSK3 signaling pathways as therapeutic targets for treating or preventing the pathogenesis of mental disorders.

## 1. Introduction

Schizophrenia (SZ) is a severe mental disorder that afflicts approximately 1% of the world’s population [1]. The manifestations of SZ are characterized by positive symptoms of delusions, hallucinations, disorganized thinking, and agitation; negative symptoms including affective flattening, alogia, and avolition; and cognitive dysfunction such as deficits in attention, learning, and memory [2]. Symptoms of SZ typically emerge during adolescence to young adulthood, which gives a window before full-blown psychosis for early intervention [3]. In patients with SZ, a prodromal period usually precedes the outbreak of psychosis [4,5,6]. Strategies for preventing the conversion from the prodromal phase to the psychotic phase are warranted [7,8]; unfortunately, trials of early intervention or preventing the onset of psychosis are still unsatisfactory [9,10,11]. This is largely due to the uncertainty of target neural circuits and the lack of ideal animal models [12,13].

The striatum composes the major part of the basal ganglia and plays key roles in motor, cognitive, and emotional functions. Abnormalities in the structure and function of the striatum have been observed in patients with SZ and are thought to be important in the pathogenies of the disease [14,15]. For example, the level of striatal D2 receptors (D2Rs) is increased in patients with SZ [16]. Patients with SZ display increased psychosis following amphetamine challenge, suggesting a hypersensitivity of D2Rs in the striatum [17]. These findings support the dopamine hypothesis of SZ, and D2Rs are the major target of anti-psychiatric medications [18,19]. In the striatum, D2R forms a protein complex with DISC1 [20], which is encoded by Disrupted-in-Schizophrenia 1 (DISC1), a susceptibility gene related to SZ [21,22,23,24,25]. DISC1 and its binding partners regulate many cellular functions such as neuronal migration, axon extension, dendritic differentiation, mitochondria motility, cargo transport, and synaptic plasticity; a variety of psychiatric phenotypes may be derived if this gene is disrupted [26,27,28].

We have characterized a *Disc1* mutant mouse line in which the full-length 100 kD DISC1 isoform is absent in homozygous knockout mice and is significantly reduced in heterozygous knockout (Het) mice [29]. We found impaired working memory and changes in neuronal properties in layer II/III of the medial prefrontal cortex of *Disc1* mutant mice, while most of the behavioral performances were comparable to the wildtype (WT) controls. We therefore suggest that our model might represent subjects in the prodromal states of SZ, in which predominant symptoms are not yet manifested [29]. In the following study, we further characterized the striatal phenotypes in Het *Disc1* mice, an SZ model of haploinsufficiency [30]. We found biochemical and morphological changes in the striatum of Het *Disc1* mice including the levels of dopamine (DA) receptors, GSK3, and PSD95 as well as the dendrites and spines of medium spiny neurons (MSNs) and density of parvalbumin (PV) neurons [30]. Notably, the effects of these changes are somewhat counterbalanced, resulting in normal locomotor activity in Het *Disc1* mice. Following the amphetamine (Amph) challenge, Het *Disc1* mice exhibited greater locomotor activity than WT controls in the open field. We then realized that the counterbalanced condition in Het *Disc1* mice might be broken by acute Amph-induced “dopamine storm”. Under such conditions, coping responses and compensatory mechanisms might not be activated in time and the striatal phenotypes of Het *Disc1* mice manifested.

We therefore considered the Het *Disc1* mice as a prodromal model of schizophrenia [30]. Environmental stressors might interact with the genetic defect of *Disc1* haploinsufficiency and disrupt the counterbalanced condition of the altered DA system in Het *Disc1* mice [30]. The DA system is sensitive to stress and the transition of SZ is thought to be associated with a dysregulated DA system, which might be aggravated by environmental or psychosocial stressors [5,19]. The response following acute treatment of Amph, impaired sensorimotor gating, may mimic the outbreak of psychosis [31]. Our Het *Disc1* mice are thus suitable for studying the conversion to psychosis and a preventive strategy.

N-acetylcysteine (NAC) is a promising multifunctional compound for treating various psychiatric and neurological disorders [32,33,34,35,36,37,38,39]. It has been proposed as a potential medication to prevent the conversion to SZ [40,41]. In the present study, we evaluated the preventive effect of chronic NAC treatment, covering the prenatal era to adulthood, on the reaction following the Amph challenge, which mimics the outbreak or conversion of psychosis in Het *Disc1* mice. Furthermore, we examined the phenotypic characteristics of the striatum in NAC-treated mice and elaborated on the possible mechanisms underlying the NAC-mediated preventive effects.

## 2. Results

N-acetylcysteine (NAC) was dissolved in the drinking water and given to female mice before pregnancy, throughout the gestation and lactation periods, and to their offering (Figure 1). No significant difference was noticed between offering mice in the WT, Het, WT-NAC, and Het-NAC groups.

### 2.1. Chronic NAC Normalized the Amph-Induced Activity in Het Disc1 Mice

We observed the response following acute Amph treatment as a sign of psychosis outbreak [31], which is augmented in Het mice [30], and tested the preventive potential of chronic NAC administration. Het *Disc1* mice exhibited enhanced amphetamine (Amph)-induced hyperactivity than the WT controls (Figure 2), as previously reported [30]. Notably, Amph-induced locomotor activity in the NAC-treated Het mice was largely reduced compared to the Het mice without NAC treatment. The traveled distance of Het-NAC was similar to that of the WT mice (Figure 2). This result demonstrates that chronic NAC treatment covering the prenatal era to adulthood effectively rescues the response to Amph challenge in Het *Disc1* mice, suggesting a preventive effect of NAC on the outbreak of psychosis.

### 2.2. Chronic NAC Adjusted the Striatum DA System in Het Disc1 Mice

It is believed that Amph-induced hyperactivity is mediated by the striatal dopamine (DA) pathway [17]. We then examined the protein level of key DA receptors, namely, dopamine D1 receptor (D1R) and dopamine D2 receptor (D2R), in the striatum using Western blot analysis (Figure 3). Without NAC treatment, the D1R level was lower in the Het mice than in the WT controls; markedly, it was increased in the NAC-treated ones (Figure 3a). D2R is closely associated with DISC1 in the striatum [20]. The level of striatal D2R is increased in patients with SZ [16], implying that D2R elevation is a pathogenic factor for SZ. A higher striatal D2R level was also noted in the Het *Disc1* mice. Significantly, the increased D2R level in the Het mice was largely reduced by NAC treatment (Figure 3b). In the WT mice, the expression of D1R and D2R was not affected by NAC treatment (Figure 3a,b).

### 2.3. Chronic NAC Increased the Expression of Striatal GSK3s in Het Disc1 Mice

Glycogen synthase kinase 3 (GSK3) signaling is important for neural development. Altered GSK3 expression and associated signaling pathways have been linked with various mental disorders including schizophrenia [42,43]. The levels of two GSK3 forms (GSK3α and GSK3β) were decreased in the Het mice, indicating a pathological condition in the striatum of these mutants. Notably, the expression of GSK3s was increased in the Het mice by NAC treatment (Figure 4a). GSK3s are known to regulate the structure of dendritic spines via the mediation of actin cytoskeleton and postsynaptic scaffold proteins such as PSD95 [44,45]. As our previous study showed reduced PSD95 in the striatum of the Het *Disc1* mice [30], we wondered whether NAC treatment could also rescue the expression of PSD95. Notably, without NAC treatment, the level of striatal PSD95 in the Het mice was lower than that in the WT controls, however, it was normalized by chronic NAC treatment (Figure 4b). In the WT mice, the expression of GSK3s and PSD95 was not affected by NAC treatment (Figure 4a,b).

### 2.4. Chronic NAC Rescued the MSN Dendritic Impairments in Het Disc1 Mice

In our Het *Disc1* mutants, along with increased striatal D2R level, we also observed a significant reduction in dendritic complexity and spine density in the striatal MSNs [30]. Since chronic NAC treatment brought the elevated striatal D2R level in the Het *Disc1* mutant mice to the WT level (Figure 3b), we wondered whether the morphology of MSNs in Het mice could also be rescued by NAC. Golgi-stained striatal MSNs were then collected and reconstructed (Figure 5a). The morphometric features of striatal MSNs were characterized. Compared to the WT group, the dendritic complexity including the number of bifurcation nodes (Figure 5b) and dendritic segments (Figure 5d) as well as the dendritic length (Figure 5e) were reduced in the Het mice. Remarkably, all of these parameters in Het mice were corrected following NAC treatment (Figure 5b–e).

The structural plasticity of the dendritic spine requires GSK3s and postsynaptic scaffold proteins such as PSD95 [44,45]. Since reduced GSK3α, GSK3β, and PSD95 levels in the Het *Disc1* mice were prevented by NAC treatment (Figure 4), we wondered whether the reduced spine density in the striatal MSNs of these mutants [30] would be rescued by NAC. To test this hypothesis, dendritic segments were collected from different dendritic orders in Golgi-stained striatal MSNs (Figure 6a). Compared with the WT group, the density of dendritic spines was largely reduced in the Het *Disc1* mice, while the reduced spine density in the Het mice was also brought back to a normal level following chronic NAC administration (Figure 6b). NAC treatment did not affect the spine density in WT mice.

### 2.5. Chronic NAC Restored Striatal PV Neurons Density in Het Disc1 Mice

In addition to the MSNs, striatal GABAergic interneurons also play important roles in regulating network function [46]. Reduced density of striatal parvalbumin (PV)-expressing interneurons has been noticed in many animal models of SZ [47,48]. We also observed decreased PV neuron density in the striatum of the Het *Disc1* mutant mice [30]. Striatal PV neurons are known to suppress MSNs, so reduced PV density might thus be associated with enhanced locomotor activity after acute Amph treatment [49]. As it has been reported that reduced PV level is associated with increased D2R expression in a rat SZ model [47], we wondered whether NAC regulates the striatal D2R level in Het mice could also adjust the PV neuron density in the striatum. We then examined the histochemical features of striatal PV-positive interneurons (Figure 7a). The density of striatal PV in Het-NAC mice was higher than in Het mice without NAC treatment (Figure 7b), indicating a protective role of NAC in the striatal PV neurons.

## 3. Discussion

In this study, we evaluated the effect of chronic NAC treatment on the striatal phenotypes of heterozygous *Disc1* mutant mice, a haploinsufficiency model of prodromal SZ. Following chronic NAC treatment, covering the prenatal era to adulthood, the Het *Disc1* mice behaved normally in the acute Amph challenge test, indicating a preventive effect on the outbreak of psychosis. Furthermore, the altered biochemical and morphological features in the striatum of these mutant mice were also normalized to the WT levels. Our results support the notion that NAC is a potential drug of choice to prevent the transition to SZ in at-risk individuals [40,41].

### 3.1. Animal Models of Prodromal SZ

In patients with SZ, a prodromal period usually precedes the outbreak of full-blown psychosis [3,4,5]. Diagnostic criteria for prodromal syndromes have been developed [50,51,52] including attenuated positive symptoms, brief intermittent psychotic syndrome, genetic risk, and recent functional decline. Attached to these criteria, the principles and limitations of the prodromal animal models are proposed [12,13,30].

In our Het *Disc1* mice, genetic risk (*Disc1* haploinsufficiency), functional decline (working memory deficit), and pathophysiological features in the striatal DA system were evident. We, therefore, considered that these mice were in the prodromal state. The interactions of these changes in the DA system were somewhat counterbalanced, resulting in their normal behavioral performances. Following the challenge of acute Amph treatment, the striatal DA level was greatly elevated so such a counterbalance in the Het *Disc1* mice is thus broken; augmented hyperlocomotor activity is then exhibited [30]. We, therefore, counted the Amph-induced DA system breaking in the Het *Disc1* mice as the outbreak of psychosis. Striatal DA elevation might further promote the transition of SZ [5,19,53]. In the current study, chronic NAC treatment that prevented the Amph-induced hyperlocomotor activity in Het *Disc1* mice is therefore considered as a preventive medication for the transition from the prodrome to SZ.

### 3.2. Multifaceted Therapeutic Potentials of NAC

NAC is known as an antioxidant and modulator of neurotransmitters including glutamate, GABA, and dopamine systems [33,37,38,39]. Here, we demonstrated that NAC is appropriate for adjusting the striatal phenotype of the Het *Disc1* mice. The enhanced Amph-induced locomotor activity in the Het *Disc1* mice indicated a dysregulated DA system. This notion is also supported by the altered expression of striatal D1R and D2R. The role of DISC1 in the regulation of D2R expression has been examined in an in vitro system [54], so we proposed a DISC1 reduction-induced D2R increase in the Het *Disc1* mic. The increase in the D2R expression may concurrently diminish the D1R expression to maintain the homeostasis of DA transmission and brain functions [30]. In the current study, chronic NAC treatment prevented the increase of D2R in the striatum. Interestingly, the decreased D1R level was also normalized.

Elevated striatal D2R density has been reported in patients with schizophrenia [16]. A striatum-specific D2R overexpression mouse model (D2R-OE mice) has been generated [55] to recapitulate the greater striatal D2R in patients with schizophrenia. Our Het *Disc1* mice shared some similarities with D2R-OE mice including typical sensorimotor gating properties, impaired working memory [9,55], and decreased MSN dendritic arbors [30,56]. Chronic NAC treatment not only normalized the dendritic arbors and the spine density of MSNs but striatal PSD95 in the Het *Disc1* mice, suggesting that their impaired glutamatergic transmission in the striatum has also been adjusted.

Striatal GABAergic inhibitory interneurons play important roles in regulating network function and behaviors [46]. In particular, PV-positive fast-spiking interneurons control the bursting of MSNs and modulate the circuit output [57]. Striatal PV neurons are known to suppress MSNs, as such, reduced PV density might thus loosen the inhibitory tone to the striatal MSNs and enhance the locomotor activity following acute Amph treatment [49]. Altered PV neurons have been noticed in various *Disc1* mutant rodent lines [48,58,59], indicating a consequential PV neuron reduction in the pathological condition of *Disc1* defects. It is still not clear how Disc1 reduction leads to PV neuron decline. Remarkably, this causal relationship seems to be unlinked to the chronic NAC treatment. Indeed, the therapeutic effects of NAC on the PV neuron deficits have been demonstrated in numerous neuropsychiatric disease animal models [60,61,62,63,64,65,66]. It has also been reported that reduced PV level is associated with increased D2R expression in a rat SZ model [47]. We thus propose that the normalization of D2R expression might be involved in the therapeutic effects of NAC on PV neuron deficits in the striatum.

NAC is the acetylated form of cysteine, which could be used for synthesizing the potent endogenous antioxidant glutathione. Mounting evidence shows augmented oxidative stress in neuropsychiatric diseases and corresponding animal models [67,68,69], and dietary supplements of an antioxidant such as NAC have shown substantial benefits [38,39,70]. Our results were in line with these reports. Oxidative stress and mitochondrial deficiency occur in several psychiatric disorders [71] and might be the result of DISC1 dysfunction [72]. NAC supplements may act downstream of DISC1 and rescue the consequence of DISC1 reduction in the Het *Disc1* mice. Furthermore, cysteine is known to modulate the cellular and extracellular glutamate levels through the cysteine-glutamate antiporter [73] and suppress the presynaptic glutamate release by metabotropic glutamate autoreceptors [74]. Enhanced glutamate release has been noticed in the mPFC neurons of *Disc1* mutant mice [29]. Chronic cysteine supplements via NAC might regulate glutamate release and assist its homeostasis in *Disc1* mutants.

Aside from the effects of NAC on the neurotransmitter systems, a recent elegant imaging study showed the involvement of microglia in a *Disc1* mutant rat model. Microglial morphology and neural microstructural features in male *Disc1* mutant rats were restored by NAC treatment, whereas chronic early-life stress reduced the therapeutic effects of NAC [75]. Since microglial activity is important for brain development, the pruning of dendritic spines as well as neuroinflammation and neuronal functions, the role of NAC in neuron–microglia interaction is warranted for further exploration [76].

### 3.3. GSK3-Mediated Protective Effects

In our current *Disc1* mutant model, elevated striatal D2R expression is evident. The binding of the DISC1 protein and D2R leads to the formation of the D2R–DISC1 complex, which plays an important role in regulating downstream signaling cascades such as the phosphorylation of GSK3 [20]. We observed decreased levels of two GSK3 isoforms (GSK3α and GSK3β) in the striatum of Het mice [30]. Altered GSK3 expression and associated signaling pathways have been linked with various mental disorders including SZ [42,43]. In fact, studies using postmortem brain samples have revealed reduced GSK3β mRNA and protein levels in SZ patients [77,78].

GSK3 signaling is important during neural development. Molecules involved in GSK3 signaling pathways regulate gene expression, protein synthesis, neurogenesis, neuronal migration, cell polarization, dendritic orientation, and structural plasticity of the dendritic spines [44,79,80,81]. The striatal MSNs in the Het *Disc1* mice showed reduced dendritic complexity and spine density, reflecting the consequence of reduced GSK3 levels [30]. The current study presented that chronic NAC treatment effectively elevated the striatal GSK3 levels and restored the dendritic complexity and spine density of MSNs in the *Disc1* Het mice. The interplay between DISC1, D2R, and GSK3 signaling during neural development is necessary for in-depth explorations.

Recent studies have revealed the roles of GSK3 in cognitive function [82]. The neurocognitive deficit is one of the core features of SZ and the treating strategy is still a major research goal in clinical psychiatry [83,84]. In our earlier study, we characterized impaired working memory in *Disc1* mutant mice. Furthermore, we also observed altered neuronal structure and synaptic transmission in the medial prefrontal cortex in these mutants, which were closely associated with their cognitive deficits [29]. We are currently examining the effects of chronic NAC treatment on the working memory function and neuronal architecture in the Disc1 mutant mice.

GSK3β has been shown to phosphorylate and inactivate 4E-BP1 and subsequently increase eIF4E-mediated protein translation [85] including BDNF [86]. BDNF is important for neuronal morphogenesis, synaptic plasticity, cognitive function, inflammation, and stress coping responses [87,88,89,90]. Our recent study showed that under the stress of sleep deprivation, Het *Disc1* mice failed to elicit the BDNF protein expression along with augmented neuroinflammation and exacerbated hippocampal neurogenesis. We associated these impaired stress coping responses with GSK3 deficiency in Het *Disc1* mice [91]. Since chronic NAC treatment normalizes the GSK3 levels in these mutants, further study of NAC on GSK3-mediated translational mechanism as well as BDNF-associated stress-coping and cognitive function might pave a way for the therapeutic strategy of neuropsychiatric disorders.

### 3.4. Limitations

There are several limitations in this present study. (1) The transition from prodrome to SZ is a complex process, so we should not directly parallel the findings in animal models and human subjects. Aside from the acute Amph challenge, we could develop chronic models such as the Amph-induced sensitization paradigm [12] to mimic the transition process. (2) The critical time window of NAC treatment has not yet been characterized. In this study, as a test of concept, NAC was given to female mice throughout the gestation and lactation period and to their offering. We should evaluate the effective period of NAC to refine the therapeutic strategy. (3) In our phenotypic characterization, only male mice were used. We should include female mice and evaluate the effects of NAC in our future study.

## 4. Materials and Methods

### 4.1. Animals

Mice were genotyped by a PCR method using a primer set (forward: 5’-GCTGTGACCTGATG GCAC-3’; Reverse: 5’-GCAAAGTCACCTCAATAACCA-3’) as previously described [29]. Breeding pairs of the male heterozygous *Disc1* mutant (Het *Disc1*) and wild-type (WT) females were set [30] and male offspring were used in this study. Mice were bred and housed in the Laboratory Animal Center of the National Taiwan University College of Medicine (NTUCM), under a 12-h light/dark cycle (lights on at 8:00) with free access to food and water. Experiments were designed under the guidelines set by the Institutional Animal Care and Use Committee (IACUC) of the NTUCM. The protocol has been reviewed and approved by the IACUC (approval number: 20180481). Efforts have been made to reduce the use of experimental animals. All animals were gently handled by an experienced experimenter to reduce their discomfort.

### 4.2. NAC Treatment

N-acetylcysteine (NAC; A7250, Sigma-Aldrich, St. Louis, MO, USA) was dissolved in the drinking water and given to the pregnant and lactating dams (2.4 g/L) and their post-weaning offspring (1 g/L) until the experiments were conducted. This protocol was chosen according to earlier studies [60,92]. In the vehicle group, regular drinking water was provided. Phenotypic examinations of the offspring were conducted at 2–3 months old.

### 4.3. Amphetamine Challenge Test

Adult male WT and Het *Disc1* mice were subjected to behavioral tests conducted between 0900 and 1700 h, as previously described [30]. To reduce the stress during the experiment, the experimenter spent time with the animals before the behavioral examination. On the day of testing, mice were habituated in the testing environment for at least 30 min. Individual mice were then placed in the center of an open field apparatus (nontransparent square acrylic box of L40 cm × W40 cm × H35 cm) and the activities were recorded for 60 min. Amph (5 mg/kg) or saline was then intraperitoneally (i.p.) injected into the mouse and its activities were recorded for another 120 min. The locomotor activities of the mice were analyzed using Topscan software (Clever System, Reston, VA, USA).

### 4.4. Western Blot Analysis

The striatal samples were collected from adult male WT and Het mice with or without NAC treatment and processed as previously described [30]. In brief, tissue samples were dissected and homogenized in RIPA buffer and centrifuged. The protein concentration in the supernatant was determined using the BCA assay (Pierce, Rockford, IL, USA). Samples were then loaded and separated by 10% SDS-PAGE and transferred onto a PVDF membrane (Immobilon-P, Millipore, Billerica, MA, USA). The membrane was blocked using 5% skim milk and incubated with primary antibodies, overnight, followed by appropriate peroxidase-conjugated secondary antibodies (see Appendix A for details). The immunoreactive bands were visualized by the chemiluminescence method (ECL, Millipore) and a UVP AutoChemiTM System (UVP Inc., Upland, CA, USA). The optical densities of the immunoreactive bands were determined with Gel-Pro analyzer software (Media Cybernetics, Silver Spring, MD, USA). The level of GAPDH was used as the internal control.

### 4.5. Immunohistochemistry

Mice were overdosed and transcardially perfused with 0.1 M PBS followed by a fixative (4% paraformaldehyde). Brain sections that were 30 μm thick were cut using a vibratome (VT1000, Leica Biosystems, Wetzlar, Germany), reacted with 1% H_2_O_2_ to block the endogenous peroxidase activity, and transferred to a PBS-based blocking solution containing 4% normal goat serum, 4% bovine serum albumin, and 0.4% Triton X-100. After blocking, sections were incubated with primary antibodies against parvalbumin (PV) overnight, followed by biotinylated secondary antibodies (see Appendix A for details) and the avidin–biotin–peroxidase complex (ABC kit, Vector Labs, Burlingame, CA, USA). Finally, sections were reacted with 3, 3′-diaminobenzidine (with 0.01% H_2_O_2_ in PBS) and mounted. The densities of the PV-positive interneurons were quantified by measuring the numbers of cells within a counting frame (250 μm × 250 μm) in the striatum.

### 4.6. Golgi Stain and Morphometric Analyses

Brain samples were kept in an impregnation solution (FD Rapid GolgiStain Kit, NeuroTechnologies, Ellicott City, MD, USA) for 4 weeks at room temperature. Impregnated samples were then washed and cut at a thickness of 150 μm using a vibratome (Leica Biosystems) and reacted with a mixture of developer and fixer solutions (FD Rapid GolgiStain Kit). Golgi-stained striatal medium spiny neurons (MSNs) were examined under a light microscope using a 20× objective lens and image stacks were obtained using the StereoInvestigator system (MicroBrightField Bioscience, Williston, VT, USA). The morphology of striatal MSNs was reconstructed using Neurolucida software (MicroBrightField Bioscience). Morphometric features including size-related and topology-related parameters were analyzed as previously described [93]. The density of MSN dendritic spines was measured at different dendritic orders using ImageJ software (NIH, Bethesda, MD, USA).

### 4.7. Statistical Analysis

All data were analyzed with SPSS (SPSS, Inc., Chicago, IL, USA). Statistical analysis was performed between groups using a two-tailed unpaired Student’s T-test, except that the time course of behaviors and monoamine levels in the Amph challenge test was analyzed by a repeated-measures ANOVA, followed by Scheff’s post hoc comparison.

## 5. Conclusions

We showed the multifaceted therapeutic potentials of NAC in neurotransmission and signaling systems in the Het *Disc1* haploinsufficiency model of SZ prodrome. In the striatum of these mutant mice, elevated D2R levels as well as reduced PV density and GSK3 expression were normalized by chronic NAC treatment. These could explain the preventive effect of NAC on Amph-induced locomotor activity in these mutants. These findings not only support the benefit of NAC as a dietary supplement for SZ prodromes, but also advance our knowledge of D2R, PV neurons, and GSK3 pathways as therapeutic targets for treating or preventing the pathogenesis of neurodevelopmental mental disorders. Future studies will emphasize the critical time window of drug administration and its underlying mechanism.

## Figures and Tables

**Figure 1 ijms-23-09419-f001:**
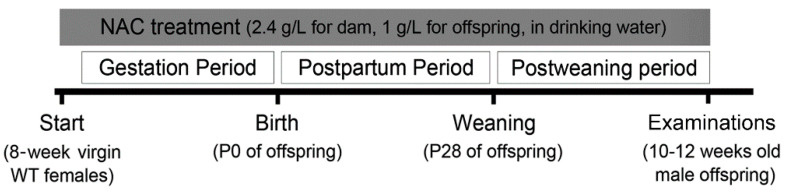
The experimental design. Drinking water with or without NAC was provided to the mice in the mating cages during the gestation and postpartum periods and in offspring cages after weaning. The day of birth was noted as postnatal day (P) 0. At P28, mice were weaned and group-reared. Phenotypic characterizations including the amphetamine challenge test, and biochemical and histological examinations were conducted at 2–3 months old.

**Figure 2 ijms-23-09419-f002:**
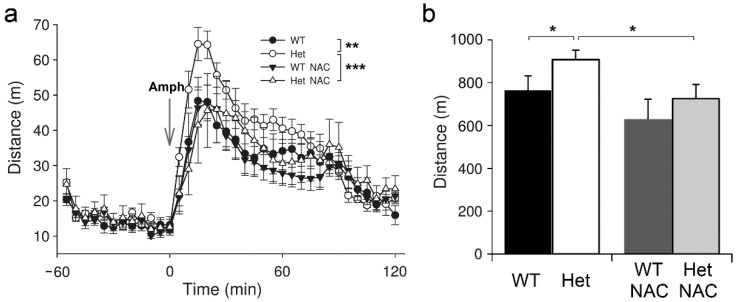
The amphetamine (Amph)-induced locomotor activity in mice. (**a**) Locomotor activity of mice was recorded and accumulated every 5 min. After 60 min of accommodation, Amph (5 mg/kg) was i.p. injected. The activity of mice was recorded for another 120 min. (**b**) The total traveled distance after Amph injection. Results are mean ± SEM. *n* = 7–11 mice per group. * *p* < 0.05; ** *p* < 0.01; *** *p* < 0.001.

**Figure 3 ijms-23-09419-f003:**
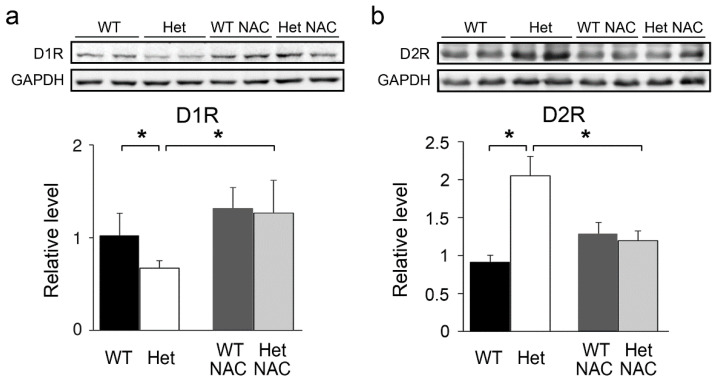
The expression of dopamine receptors in the striatum. The levels of striatal dopamine D1 receptor (D1R) and dopamine D2 receptor 2 (D2R) were examined using a Western blotting assay. GAPDH was used as the internal control. The expression of D1R was lower in the Het *Disc1* mice while the decreased D1R level was adjusted by NAC treatment (**a**). The level of D2R was higher in the Het *Disc1* mice while the increased D2R expression was reduced by NAC treatment (**b**). Results are mean ± SEM. *n* = 6–10 mice per group. * *p* < 0.05.

**Figure 4 ijms-23-09419-f004:**
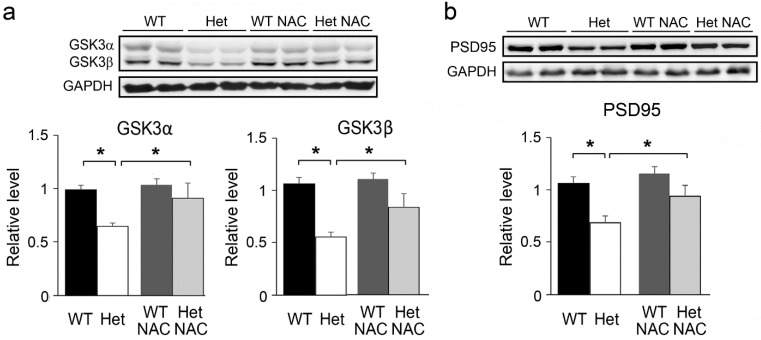
The expression of GSK3 and PSD95 in the striatum. (**a**) Expression of glycogen synthase kinase 3 (GSK3) in the striatum. The levels of GSK3α and GSK3β in the Het Disc1 mice were lower than those in WT, while the levels were increased by NAC treatment. (**b**) Expression of PSD95 in the striatum. The expression of PSD95 was lower in the Het *Disc1* mice while the decreased PSD95 level was adjusted by NAC treatment. GAPDH was used as the internal control. The results are the mean ± SEM. *n* = 6–10 mice per group. * *p* < 0.05.

**Figure 5 ijms-23-09419-f005:**
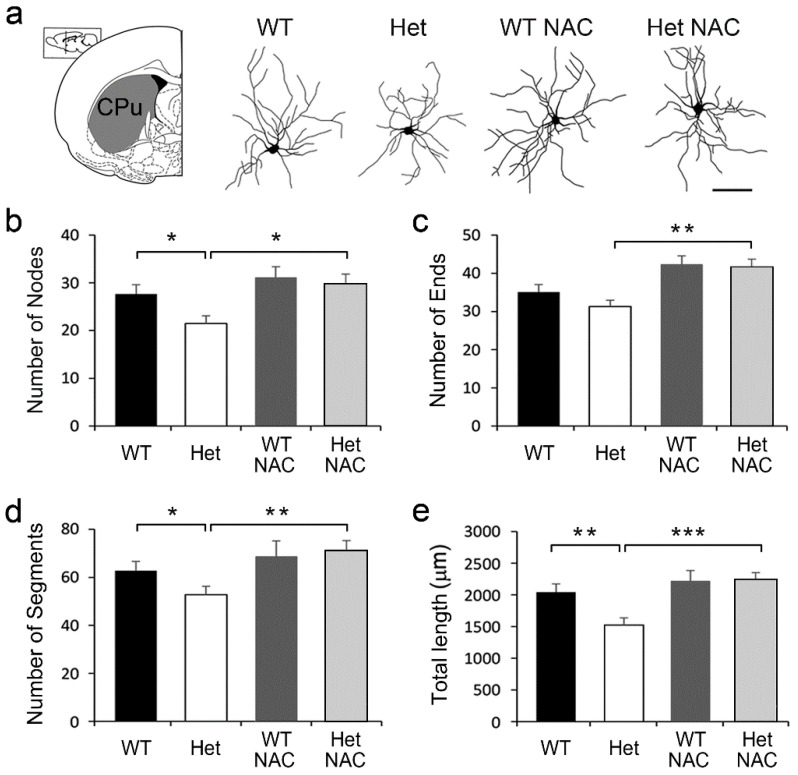
The morphological features of the striatal medium spiny neurons (MSNs). (**a**) Golgi-stained MSNs in the striatum (CPu) were collected and reconstructed. (**b**–**e**) The morphometric features including the number of branching nodes (**b**), terminal ends (**c**), dendritic segments (**d**), and total dendritic length (**e**) were measured. The results are the mean ± SEM. Data were collected from 30–40 MSNs from 4–6 mice per group. * *p* < 0.05; ** *p* < 0.01; *** *p* < 0.001.

**Figure 6 ijms-23-09419-f006:**
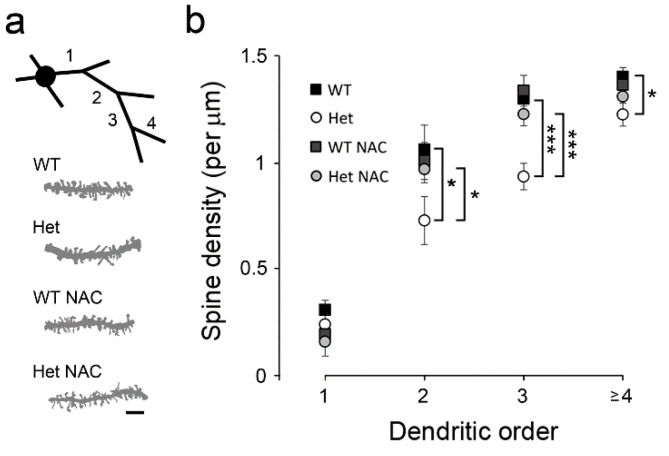
The dendritic spine density of the striatal MSNs. Dendritic segments of different orders were collected from Golgi-stained striatal MSCs (**a**). The densities of dendritic spines were counted (**b**). Bar = 5 μm in A. Data are the mean ± SEM. Data were obtained from 20–30 MSNs from 4–6 mice per group. * *p* < 0.05; *** *p* < 0.001.

**Figure 7 ijms-23-09419-f007:**
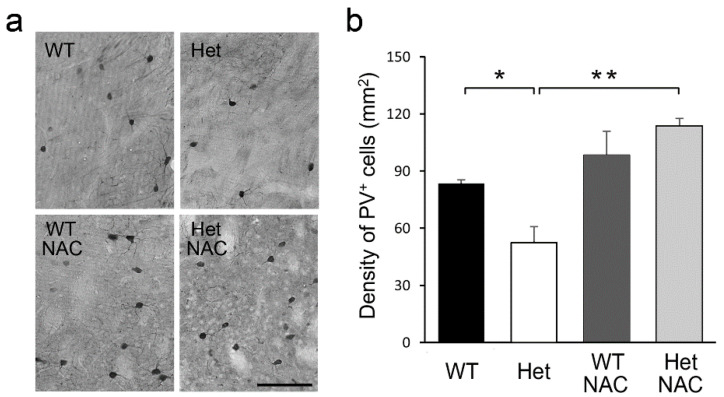
The density of PV interneurons in the striatum. (**a**) Immunolabeled striatal PV-positive interneurons. (**b**) The density of striatal PV-positive neurons was lower in the Het Disc1 mutant mice and increased by NAC treatment. Bar = 100 μm. Results are the mean ± SEM. N = 4–6 mice per group. * *p* < 0.05; ** *p* < 0.01.

## Data Availability

Data are available on request from the authors.

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
