# Peer review of "Chronic N-Acetylcysteine Treatment Prevents Amphetamine-Induced Hyperactivity in Heterozygous *Disc1* Mutant Mice, a Putative Prodromal Schizophrenia Animal Model"

_ijms, 2022, doi:10.3390/ijms23169419_

Round 1

Reviewer 1 Report

Lai and collaborators look at the effect of a chronic NAC treatment in a mouse model of schizophrenia prodrome. My main concern is that the rationale for considering these mice model as being in the prodromal stage of schizophrenia is quite narrow and it is difficult to relate it to the complex reality of the illness. In the discussion, the authors talk about 'a haploinsufficiency model of prodromal SZ', this should be specified already in the introduction.

They observed a protective effect of NAC to Amphetamine-induced hyperactivity.

The methodology seems adequate and the results are interesting and promising, but the context needs to be softened and direct parallels between animal models and the expression of the illness in humans should be avoided.

Introduction:

"In the striatum, D2Rs are closely associated with 60 DISC1 [20] which is encoded by Disrupted-in-Schizophrenia 1 (DISC1), a susceptibility 61 gene related to SZ [21-25]." The term 'associate' is not specific enough here. Co-located?

Is DISC1 itself elevated or reduced in schizophrenia?

What are the neurobiological consequences of an elevated D2R-DISC1 complex? Would breaking down the D2r-DISC1 complex pharmaceutically have an effect on the normal functioning of the complex?

"We have reported a Disc1 mutant mouse line" consider replacing 'reported' by engineered? Characterised?

What is the purpose of the amphetamine challenge?

"We, therefore, considered these mice as a prodromal model of schizophrenia [30]." this is a big jump. This would be true if the mice when on to exhibit behavioural abnormalities later in life. Also, impaired sensorimotor gating may be a trigger for psychosis in a subset of patients, but not in all. The authors seems to over simplify the reality.

Results

The first paragraph of results and figure 1 should be part of the methods section. E.g. How was NAC dissolved in water? Number and species of mice studied? Amph challenge protocol? Methods used for all measures, etc.

This section should only hold results from the current study. Results from other studies should be presented in the introduction to support the rationale or in the discussion to compare/explain the results obtained against the literature.

Discussion

First two sentence are not needed.

Considering that NAC treatment was given chronically since gestation, how do the authors

Materials and Methods

How was NAC dissolved in water?

Conclusion

"we showed the multifaceted therapeutic potentials of NAC in neurotransmission and signalling systems" It would be useful here, or at the start of the discussion to summarise, in a sentence or two, what these effects of NAC are. This summary is somewhat present in the Abstract, but the direction of the changes observed is missing.

Author Response

To Reviewer 1

Comments and Suggestions for Authors

Lai and collaborators look at the effect of a chronic NAC treatment in a mouse model of schizophrenia prodrome. My main concern is that the rationale for considering these mice models as being in the prodromal stage of schizophrenia is quite narrow and it is difficult to relate it to the complex reality of the illness. In the discussion, the authors talk about 'a haploinsufficiency model of prodromal SZ', this should be specified already in the introduction.

Response >> Thanks for the comment. We elaborated on the rationale for considering our model as a haploinsufficiency model of prodromal schizophrenia (SZ) in the Introduction of our revision.

They observed a protective effect of NAC to Amphetamine-induced hyperactivity. The methodology seems adequate and the results are interesting and promising, but the context needs to be softened and direct parallels between animal models and the expression of the illness in humans should be avoided.

Response >> Thanks for the comment. We softened our tone and avoided the direct parallel between animal models and SZ patients in the Discussion.

"In the striatum, D2Rs are closely associated with 60 DISC1 [20] which is encoded by Disrupted-in-Schizophrenia 1 (DISC1), a susceptibility 61 gene related to SZ [21-25]." The term 'associate' is not specific enough here. Co-located?

Response >> Thanks for the comment. We rephrased the sentence as “In the striatum, D2R forms a protein complex with DISC1 which is encoded by Disrupted-in-Schizophrenia 1 (DISC1), a susceptibility gene related to SZ”.

Is DISC1 itself elevated or reduced in schizophrenia?

Response >> The levels of DISC1 in the peripheral blood and postmortem brain tissue of SZ patients were examined, however, the findings are inconsistent. For example, reduced DISC1 protein and mRNA levels in the peripheral blood of patients with SZ have been reported (Trossbach et al., 2014; Chen et al., 2022); however, Fu et al. found higher DISC1 gene expression in the peripheral blood of SZ patients (Fu et al., 2020). Reduced DISC1 protein levels have been noted in the prefrontal cortex of SZ patients (Ratta-apha et al., 2013) while no difference in DISC1 mRNA levels between patients and healthy controls has also been reported (Rastogi et al., 2009). These inconsistencies might be due to the numerous variants and isoforms of the DISC1 gene and protein. 

What are the neurobiological consequences of an elevated D2R-DISC1 complex? Would breaking down the D2R-DISC1 complex pharmaceutically have an effect on the normal functioning of the complex?

Response >> The effects of assembly and disassembly of the D2R-DISC1 complex were not examined and discussed in our study. We could only reply to the questions raised by the reviewer based on the reports of Su et al. (2014). These authors proposed a model that in the presence of DISC1, the D2R-DISC1 complex leads to the activation of GSK-3 signaling while in the absence of DISC1, D2R tends to be internalized. Under physiological circumstances, these two conditions are in equilibrium. Elevated levels of the D2R-DISC1 complex, which were observed in postmortem brain tissue from SZ patients, would lead to over-activation of the GSK3 pathway and a relative decrease in D2R internalization (Su et al., 2014). These authors also developed an interfering peptide (D2pep) that disrupts the D2R-DISC1 complex. D2pep has no effect on D2R-mediated cAMP accumulation, locomotor activity, and sensorimotor gating in mice. However, the effects of D2Rpep on other D2R-related signaling pathways and functions can not be ruled out (Su et al., 2014). In order to focus our study on DISC1, we removed the D2R-DISC1 complex part in our Introduction.

"We have reported a Disc1 mutant mouse line" consider replacing 'reported' by engineered? Characterized?

Response >> Thanks for the comment. We replaced “reported” with “characterized”. 

What is the purpose of the amphetamine challenge?

Response >> Although the amphetamine (Amph) challenge is commonly used in characterizing animal models of SZ, in our present study, we used this challenge to induce a “dopamine storm” in order to disrupt the counterbalanced condition of the altered dopamine system in Disc1 Het mice.

In our previous study, we found biochemical and morphological changes in the striatum of Het Disc1 mice including the levels of dopamine receptors, GSK3 and PSD95, as well as the dendrites and spines of MSNs and density of PV neurons (Baskaran et al., 2020). Notably, the effects of these changes are somewhat counterbalanced, resulting in normal locomotor activity in Het Disc1 mice. Following Amph, Het Disc1 mice exhibited greater locomotor activity than WT controls in the open field, we then realized the counterbalanced condition in Het Disc1 mice might be broken by this acute treatment, and the striatal phenotypes of Het Disc1 mice are manifested. We added this notion in the Introduction of our revised version.

"We, therefore, considered these mice as a prodromal model of schizophrenia [30]." this is a big jump. This would be true if the mice when on to exhibit behavioral abnormalities later in life. Also, impaired sensorimotor gating may be a trigger for psychosis in a subset of patients, but not in all. The authors seem to oversimplify the reality.

Response >> Animal models of SZ can only mimic some aspects of the SZ symptoms, the authors, therefore, agreed with the reviewer that we seem to oversimplify the reality. However, our animal model could still make a substantial contribution to the study of preventive strategies. In our earlier study (Juan et al., 2014), we found impaired working memory and changes in neuronal properties in layer II/III of the medial prefrontal cortex of Disc1 mutant mice, while most of the behavioral performances are comparable to WT controls. We, therefore, suggested that “our model might represent subjects in the prodromal states of psychosis, in which predominant symptoms are not yet manifested” (Juan et al., 2014). In the following study, we elaborated on the phenotypes in Disc1 Het mice and found greater Amph-induced locomotor activity compared with WT controls (Baskaran et al., 2020). The findings in our Het Disc1 mutant mice support the haploinsufficiency model of SZ. We rephrased this section in the Introduction of our reversion.

Results

The first paragraph of the results and figure 1 should be part of the methods section. E.g. How was NAC dissolved in water? Number and species of mice studied? Amph challenge protocol? Methods used for all measures, etc.

Response >> Thanks for the comment. Because of the style of IJMS, the Introduction is followed by Results, we hoped this layout gives a clear experimental design for the readers.

This section should only hold results from the current study. Results from other studies should be presented in the introduction to support the rationale or in the discussion to compare/explain the results obtained against the literature.

Response >> Thanks for the suggestion. We removed some citations and results from other studies in this section.

Discussion

The first two sentences are not needed.

Response >> Thanks for the suggestion. We removed the first two sentences in the first paragraph of the Discussion.

Considering that NAC treatment was given chronically since gestation, how do the authors

Response >> Sorry, the question is not complete, we are unable to answer. 

Materials and Methods

How was NAC dissolved in water?

Response >> NAC was completely dissolved in the drinking water.

Conclusion

"we showed the multifaceted therapeutic potentials of NAC in neurotransmission and signaling systems" It would be useful here, or at the start of the discussion to summarise, in a sentence or two, what these effects of NAC are. This summary is somewhat present in the Abstract, but the direction of the changes observed is missing.

Response >> Thanks for the comment. We rephrased the Discussion, Conclusion, and Abstract accordingly.

References

Baskaran R, Lai CC, Li WY, Tuan LH, Wang CC, Lee LJ, Liu CM, Hwu HG, Lee LJ. Characterization of striatal phenotypes in heterozygous Disc1 mutant mice, a model of haploinsufficiency. J Comp Neurol. 2020 528(7):1157-1172. doi: 10.1002/cne.24813.

Chen YM, Lin CH, Lane HY. Distinctively lower DISC1 mRNA levels in patients with schizophrenia, especially in those with higher positive, negative, and depressive symptoms. Pharmacol Biochem Behav. 2022 213:173335. doi: 10.1016/j.pbb.2022.173335.

Fu X, Zhang G, Liu Y, Zhang L, Zhang F, Zhou C. Altered expression of the DISC1 gene in peripheral blood of patients with schizophrenia. BMC Med Genet. 2020 Oct 2;21(1):194. doi: 10.1186/s12881-020-01132-9..

Juan LW, Liao CC, Lai WS, Chang CY, Pei JC, Wong WR, Liu CM, Hwu HG, Lee LJ. Phenotypic characterization of C57BL/6J mice carrying the Disc1 gene from the 129S6/SvEv strain. Brain Struct Funct. 2014 219(4):1417-31. doi: 10.1007/ s00429-013-0577-8.

Rastogi A, Zai C, Likhodi O, Kennedy JL, Wong AH. Genetic association and post-mortem brain mRNA analysis of DISC1 and related genes in schizophrenia. Schizophr Res. 2009 114(1-3):39-49. doi: 10.1016/j.schres.2009.06.019.

Ratta-Apha W, Hishimoto A, Mouri K, Shiroiwa K, Sasada T, Yoshida M, Supriyanto I, Ueno Y, Asano M, Shirakawa O, Togashi H, Takai Y, Sora I. Association analysis of the DISC1 gene with schizophrenia in the Japanese population and DISC1 immunoreactivity in the postmortem brain. Neurosci Res. 2013 77(4):222-7. doi: 10.1016/j.neures.2013. 08.010.

Su P, Li S, Chen S, Lipina TV, Wang M, Lai TK, Lee FH, Zhang H, Zhai D, Ferguson SS, Nobrega JN, Wong AH, Roder JC, Fletcher PJ, Liu F. A dopamine D2 receptor-DISC1 protein complex may contribute to antipsychotic-like effects. Neuron. 2014 Dec 17;84(6):1302-16. doi: 10.1016/j.neuron.2014.11.007.

Trossbach SV, Fehsel K, Henning U, Winterer G, Luckhaus C, Schäble S, Silva MA, Korth C. Peripheral DISC1 protein levels as a trait marker for schizophrenia and modulating effects of nicotine. Behav Brain Res. 2014 Dec 15;275:176-82. doi: 10.1016/j.bbr.2014.08.064.

Reviewer 2 Report

The present study investigates the protective effect of chronic N-acetylcysteine treatment on the development of amphetamine-induced hyperactivity in heterozygous Disc1 mice. These mice can be considered a putative model of prodromal-phase schizophrenia, and the amphetamine challenge can be considered a model of psychotic onset.

The study is timely and interesting, investigating a topic clinical relevance that requires further study. The methods are detailed and would allow study replication. The statistical analyses are appropriate. The results are interesting. The manuscript is very well-written and easy to read.

Overall, the present paper represents a worthy contribution to the research field.

However, some revisions could further improve the clarity and the interest for the reader of the manuscript.

Discussion:

-Cognitive impairment represents one of the core features of schizophrenia spectrum disorders, and treating cognitive deficits represents a major current research goal (see Vita A et al. Effectiveness, Core Elements, and Moderators of Response of Cognitive Remediation for Schizophrenia: A Systematic Review and Meta-analysis of Randomized Clinical Trials. See Veselinović T and Neuner I. Progress and Pitfalls in Developing Agents to Treat Neurocognitive Deficits Associated with Schizophrenia. CNS Drugs. 2022; 36:819–858. doi.org/10.1007/s40263-022-00935-z). A comment on this issue related to the results of the present study could be of interest.

-A paragraph dedicated to the strength and limitations of the study should be implemented in the discussion section. In particular, while this is briefly mentioned in the manuscript, a more extensive discussion should be provided regarding the limited generalizability of the present interesting results to human psychopathology and clinical practice.

-More discussion on future research perspectives could also be provided.

Author Response

To Reviewer 2

Comments and Suggestions for Authors

The present study investigates the protective effect of chronic N-acetylcysteine treatment on the development of amphetamine-induced hyperactivity in heterozygous Disc1 mice. These mice can be considered a putative model of prodromal-phase schizophrenia, and the amphetamine challenge can be considered a model of psychotic onset. The study is timely and interesting, investigating a topic clinical relevance that requires further study. The methods are detailed and would allow study replication. The statistical analyses are appropriate. The results are interesting. The manuscript is very well-written and easy to read. Overall, the present paper represents a worthy contribution to the research field. However, some revisions could further improve the clarity and the interest for the reader of the manuscript.

Response >> Thanks for the encouragement.

Discussion:

-Cognitive impairment represents one of the core features of schizophrenia spectrum disorders, and treating cognitive deficits represents a major current research goal (see Vita A et al. Effectiveness, Core Elements, and Moderators of Response of Cognitive Remediation for Schizophrenia: A Systematic Review and Meta-analysis of Randomized Clinical Trials. See Veselinović T and Neuner I. Progress and Pitfalls in Developing Agents to Treat Neurocognitive Deficits Associated with Schizophrenia. CNS Drugs. 2022; 36:819–858. doi.org/10.1007/s40263-022-00935-z). A comment on this issue related to the results of the present study could be of interest.

Response >> Thanks for the suggestion. In our earlier study, we characterized impaired working memory in Disc1 mutant mice. Further, we also observed altered neuronal structure and synaptic transmission in the medial prefrontal cortex in these mutants which were closely associated with their cognitive deficits (Juan et al., 2014). We are currently examining the effects of chronic NAC treatment on the working memory function and neuronal architecture in Disc1 mutant mice. We added the above-mentioned references and current work in the Discussion of our revision.     

-A paragraph dedicated to the strength and limitations of the study should be implemented in the discussion section. In particular, while this is briefly mentioned in the manuscript, a more extensive discussion should be provided regarding the limited generalizability of the present interesting results to human psychopathology and clinical practice.

Response >> Thanks for the suggestion. We added the limitations of the study in the revised Discussion.  

-More discussion on future research perspectives could also be provided.

Response >> Thanks for the suggestion. We included future research perspectives in our revised Discussion.